# Spatial Distribution and Accessibility Measurements for Elderly Day Care Centers in China's Urban Built-up Area: The Case of Tianjin Nankai District

Da Wan [1,2], Hui Liu [1], Jiaxing Guo [1,3], Lian Guo [1], Dongchen Qi [1], Sheng Zhang [1,*], Pengbo Li [1] and Hiroatsu Fukuda [2]

1   School of Architecture, Tianjin Chengjian University, Tianjin 300380, China
2   Department of Architecture, Faculty of Environmental Engineering, The University of Kitakyushu, Kitakyushu 808-0135, Japan
3   Tianjin Architecture Design Institute Co., Ltd., Tianjin 300074, China
*   Correspondence: zs_057@163.com

**Abstract:** The elderly community day care model is an emergent solution to the aging problem influenced by the Eastern perspective of family in China. Due to the structural problem of spatial disorder in most of China's urban built-up areas, the planning and construction of elderly day care centers (EDCCs) is facing great challenges. This study aims to comprehensively compare the spatial distribution and accessibility measurement methods for elderly residents and EDCCs in typical Chinese urban built-up areas based on the accessibility theory and spatial analysis methods from the community living circle perspective. The results show that different spatial distribution analysis methods have their own emphases and limitations, requiring comprehensive application in practice. The potential model method is most suitable for the accessibility measurement in this scenario. The threshold setting of service distance for the urban built-up areas public service facilities in the current Chinese standard needs to be further optimized. The existing EDCCs suffer from serious quantity deficiencies and misplaced supplies in the region. These findings can reveal the EDCCs distribution characteristics of typical Chinese urban built-up areas and provide new insights for urban planners and policy makers who are assessing the equity and efficiency of public service facilities.

**Keywords:** elderly care; public facility planning; spatial allocation; service distance; ArcGIS

## 1. Introduction

According to the report of the Ministry of Civil Affairs of China in 2020, there are 264 million people aged 60 and over in China, accounting for 18.7% of the total population. The population aged 65 and over is 191 million, accounting for 13.5%, which is expected to reach 14% in 2025. The increasing number of elderly groups, such as empty nesters, singletons, and semi-disabled will pose a significant challenge to China's elderly service system. Many outstanding elderly care systems can be observed around the world, especially in developed countries. Examples include the foster care programs and self-helping elderly living communities in Sweden, Belgium, and Denmark; the self and mutually supporting elderly care services in the United States; the family retirement insurance program in Singapore; the community-based integrated care system in Japan; and the community care and assistance services and residential care services for the elderly in Hong Kong [1,2].

In China, despite the growth of smaller and more nucleated families due to urbanization, the traditional family-based concept still leans toward a "family-centered" elderly care model, with a certain degree of clannishness and exclusivity. To address these challenges, the community elderly day care mode allows the elderly who are independent or semi-independent to access day care services, such as meals, personal care, physical examinations, and social recreation through the community, with the family still as the core.

This is an intermediate model between home care and institutional care, preserving the traditional form of home care while fully utilizing personal, family, community, and social resources to provide close and convenient services to the elderly. This not only satisfies the traditional psychological and material demands of the elderly, but also reduces the daily care load for their children.

The previous community planning system of the "per 1000 population model" in China focused on supply-side planning. It measured the floor area and land area of each public service facility per 1000 residents as the control index for urban public facilities. This is a top-down allocation process from the supply-side perspective. The implementation path is a one-way path from supply planning to users, without considering the feedback mechanism from users to the supply side. However, the growing multi-level demands of residents are not easily met by this system. The mismatch between the supply and demand structure is a critical issue. To cope with it, the latest Standard for Urban Residential Area Planning and Design (GB 50180-2018) [3] (hereinafter referred to as "the Standard") changed the classification model of residential areas based on the number of residents to the community living circle classified by pedestrian scale. The theoretical basis of this model is the spatial accessibility, which was first introduced by Hansen in 1959 as the magnitude of the interactive impact of each node in the road networks [4].

In recent years, with the development of computer technology and social science research, the study on accessibility has been gradually intensified. Several scholars have reviewed accessibility research in terms of road networks' evaluation, transportation efficiency comparison, and geographic region research [5–9]. Regarding the spatial distribution of public service facilities, the main research objects involve high demand for physical proximity of urban public service facilities, such as schools, hospitals, and greenspaces [10–15]. Some studies focused on the accessibility of low-income workers, the elderly, and disabled people [16–18]. Currently, spatial accessibility is one of the most effective tools to measure the spatial allocation rationality of public service resources. It is widely used in the evaluation of emergency services, transportation, education, medical care, and other public facilities arrangements [19–23]. Based on these studies, the pedestrian scale community classification model developed in China guides the distribution of service facilities with accessibility to meet the requirements of residents who can reach the corresponding facilities within an appropriate walking timeframe. Public service facilities will progressively develop a distribution strategy that is constrained from both the supply side and the demand side.

However, due to social and historical conditions, the early construction of urban areas failed to anticipate the rapid aging of China's population. Therefore, the number of community elderly facilities allocated in the early urban construction was few, and the distribution was not discussed in a scientific manner. There is still limited study on the spatial distribution and accessibility of EDCCs in China's urban built-up areas. In addition, how to carry out an effective arrangement of elderly facilities in built-up areas with high density and how to promote the construction of such facilities are still two of the difficulties to the researchers.

This study aims to provide a theoretical basis and methodological support for the planning and renovation of EDCCs in China's urban built-up areas by examining the spatial distribution and accessibility measurement methods. Firstly, we investigated the EDCCs for the elderly in Nankai District, Tianjin, and obtained the basic construction data of EDCCs and the data of residences in Nankai District and the surrounding communities. After that, we combined the road network data to construct the database of "EDCCs—residences". Then, the standard deviation ellipse (SDE), spatial autocorrelation (SA), average nearest neighbor (ANN), kernel density estimation (KDE), and buffer zone methods were used to comprehensively evaluate the spatial distribution characteristics of EDCCs in the region. The accessibility was analyzed and compared by the proportional method, the nearest distance method, the two-step floating catchment area method (2SFCA), the inverted two-step floating catchment area method (i2SFCA), and the potential model method. The

results show the emphases of different spatial distribution algorithms and their limitations. Regarding 2SFCA and the potential model, the latter demonstrated more advantages for the analysis of this scenario. The threshold setting of public facilities' service distances for the urban built-up areas in the current standard requires further exploration. The analysis results also confirmed that the number of existing EDCCs in Nankai District is severely small, the distribution is random, and a spatially mismatch in supplies with the distribution of the elderly population exists. Finally, the principles and strategies for planning and optimization of EDCC are presented based on the analysis (Figure 1). These findings of this study may provide new insights for urban planners and policy makers in assessing the equity and efficiency of the accessibility and spatial distribution of public service facilities.

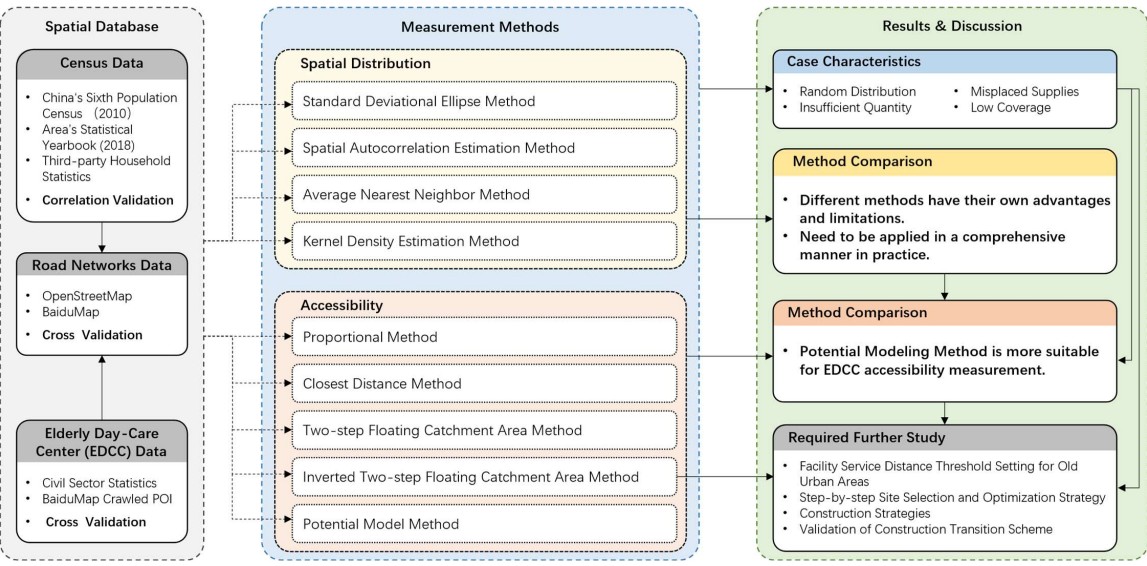

**Figure 1.** Study framework.

## 2. Materials and Methods

### 2.1. Study Area

According to the 6th Census in 2010, Nankai District has a population of 1,018,196, of which 16.78% (170,885 people) are 60 years old and above, which is similar to the national data. The district has existed since the 14th century. It includes 12 sub-districts and 171 communities (Table 1). It was selected as the second batch of national experimental areas for reforming home and community-based elderly care services in 2017 and is typical case of elderly service development recommended by the Ministry of Civil Affairs (Figure 2).

**Table 1.** Basic information of Nankai sub-districts.

| Name | Area (km²) [a] | Number of Community [b] | Aging Rate [b] (Aged 60 and Over, %) | Elderly Population Density (Persons/km²) |
|---|---|---|---|---|
| Gulou | 2.117 | 9 | 7.55 | 1061 |
| Xingnan | 1.7 | 9 | 16.9 | 5102 |
| Guangkai | 1.7 | 13 | 15.1 | 6458 |
| Changhong | 2.635 | 13 | 14.09 | 3930 |
| Xiangyanglu | 10 | 19 | 16.72 | 5962 |
| Jialingdao | 3.672 | 16 | 19.29 | 5048 |
| Wanxing | 3.54 | 22 | 18.66 | 7697 |
| Xuefu | 4.7 | 14 | 14.23 | 3384 |
| Shuishanggongyuan | 5.633 | 10 | 13.92 | 1717 |
| Wangdingdi | 5.016 | 23 | 21.19 | 5542 |
| Tiyuzhongxin | 6.467 | 12 | 14.08 | 1601 |
| Huayuan | 1.92 | 11 | 19.79 | 5049 |

Data resource: [a] Nankai District Statistical Yearbook 2018; [b] 6th Census of China.

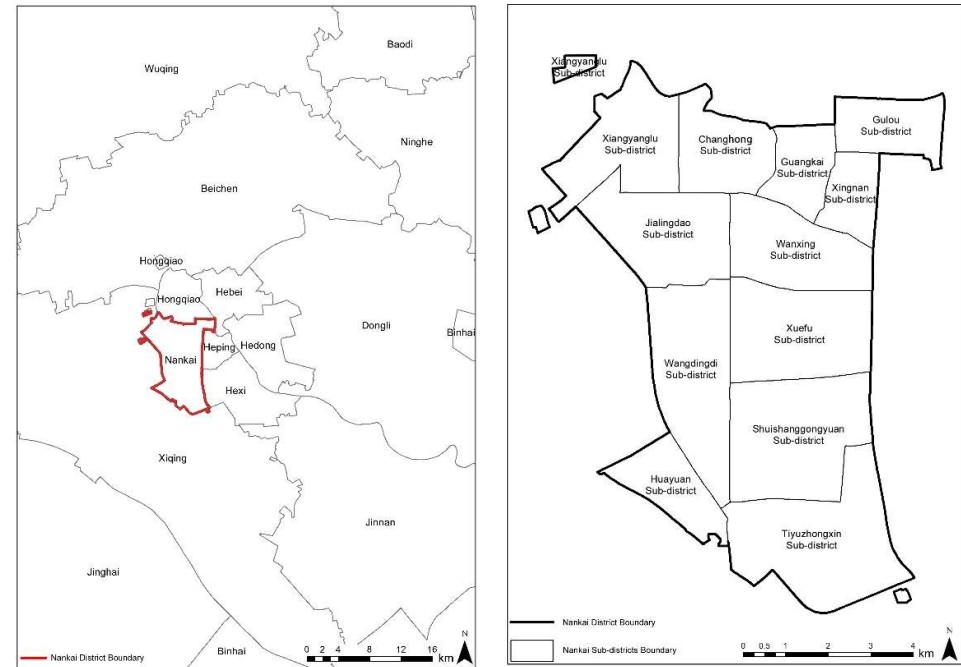

**Figure 2.** Study area. Left: Location of Nankai District. Right: Sub-districts of Nankai District.

*2.2. Related Standard*

The Standard, implemented in 2018, is the main authority for the allocation of community public service facilities. There is a fundamental change in the classification of residential areas. From the classifying of residential groups, blocks, and areas based on the number of households, it has changed to the living circles system based on the pedestrian scale. This change is combined with residents' travel behavior to guide the arrangement of facilities (Table 2). The Standard stipulates that a 5 min pedestrian scale neighborhood should include EDCCs, and the services' distances should not exceed 300 m.

**Table 2.** Classification model of residential areas.

| Item | 15-Min Pedestrian-Scale Neighborhood | 10-Min Pedestrian-Scale Neighborhood | 5-Min Pedestrian-Scale Neighborhood | Neighborhood Block |
|---|---|---|---|---|
| Walking Distance (m) | 800~1000 | 500 | 300 | - |
| Resident Population | 50,000~100,000 | 15,000~25,000 | 5000~12,000 | 1000~3000 |
| Number of Households | 17,000~32,000 | 5000~8000 | 1500~4000 | 300~1000 |

Data resource: Urban Residential Area Planning and Design Standard (GB 50180-2018).

*2.3. Data Collection and Spatial Database*

1.　Road Networks

Urban road networks are important to know about when performing spatial distribution and accessibility analysis. For this study, road network data were provided by Urban Data Party (Tianjin Road Networks, based on Baidu Maps) as the base map and validated with Open Street Map (OpenStreetMap).

2.　Elderly Population

The source of the sub-district population data was the 6th China Census (2010), applied through the Nankai District Information Disclosure Platform (Table 3).

**Table 3.** Elderly populations of Nankai sub-districts in 2010.

| Name | Total Population | Aged 65 and Over | Aged 60 and Over | Aging Rate (%) |
|---|---|---|---|---|
| Gulou | 28,788 | 1264 | 2174 | 7.55% |
| Xingnan | 43,744 | 4922 | 7392 | 16.90% |
| Guangkai | 73,299 | 7357 | 11,071 | 15.10% |
| Changhong | 73,346 | 7298 | 10,337 | 14.09% |
| Xiangyanglu | 145,263 | 16,425 | 24,293 | 16.72% |
| Jialingdao | 102,405 | 13,767 | 19,757 | 19.29% |
| Wanxing | 140,684 | 18,900 | 26,253 | 18.66% |
| Xuefu | 103,319 | 11,137 | 14,699 | 14.23% |
| Shuishanggongyuan | 64,853 | 6389 | 9029 | 13.92% |
| Wangdingdi | 122,599 | 18,368 | 25,975 | 21.19% |
| Tiyuzhongxin | 66,948 | 6292 | 9429 | 14.08% |
| Huayuan | 52,948 | 7304 | 10,476 | 19.79% |

Data resource: 6th Census of China.

## 3. Population Variables

It has been more than a decade since the 6th Census in China. To ensure the timeliness of the study, the total number of households in each sub-district was counted using the ArcGIS overlay analysis tool, and a Pearson correlation test was conducted with the total population and the number of elderly populations, respectively. By means of data crawling, the household data of Nankai District was obtained from the Lianjia (a Chinese real-estate brokerage company: lianjia.com, accessed on 23 July 2022). Data from five adjacent districts were used for joint analysis to avoid boundary effects. The results show that the correlation coefficients are 0.777 and 0.796, respectively, and both are significantly correlated at the 0.01 level (2-tailed). This proves that the elderly population shows a strong correlation with the number of households in the area (Table 4). Combining the above correlation analysis and the household capacity criteria of the community living circle, the number of households was used as a population variable in this study for analysis and calculation.

**Table 4.** Correlation test between the number of households and population in 12 sub-districts.

| | Correlation | Population | Population Aged 60 and Over |
|---|---|---|---|
| Households | Pearson Correlation | 0.777 | 0.796 |
| | Sig. (2-tailed) | 0.003 [1] | 0.002 [1] |

[1] At the 0.01 level (2-tailed), the correlation is significant.

## 4. EDCC Data

The EDCC data were retrieved from the Civil Affairs Department of Nankai District and compiled by online and offline investigation. The data include the locating sub-district, name, detailed address, building area, number of staff, number of beds, type of services, etc. The geographical coordinates were crawled by Baidu Maps API (lbsyun.baidu.com/jsdemo.htm#wAddressParseGroup, accessed on 23 July 2022) and then rectified.

## 5. Spatial Database

The spatial database was constructed by ArcGIS. Only walking-distance EDCCs were considered in this study. Therefore, the original design speed of the road was not considered, and the traffic cost was calculated by the length of the road network. The population data were set as surface elements, containing the total population, the elderly population aged over 60, and the aging rate of the sub-district.

### 2.4. Spatial Distribution Analysis Methods

Number of facilities and density data can be used in a preliminary statistical analysis of the arrangement but fail to reflect the spatial distribution characteristics. Therefore, this study conducted comparative spatial distribution analysis by using the SDE, SA, an ANN, a KDE, and the buffer zone method in ArcGIS. These could identify the relationship between EDCCs and elderly distribution from various aspects.

1.    Standard deviation ellipse method (SDE)

SDE can analyze the center of spatial distribution and the trend of distribution. The long axis of the ellipse indicates the direction with the most spatially distributed points, and the short axis indicates the direction with the least spatially distributed points [24,25]. The center of the ellipse is the average coordinates of all points.

In this study, SDE was conducted separately for Nankai District residences weighted with the elderly population, and Nankai District EDCCs with the facility area as the weighting value. The standard deviation level was set to one, and the proportion of data inclusion was 68%.

2.    Spatial autocorrelation method (SA)

SA (Global Moran's I) is an important research method in spatial statistics to study the phenomenon of objects' spatial distribution, or whether there is a correlation between an attribute of the object and its spatial distribution [26,27]. It is mainly used to test whether a certain attribute of the object has a high-to-high adjacency or high-to-low misallocation.

In this study, the correlation between the spatial distribution of EDCCs and their area of location was analyzed by ArcGIS using the global Moran's I (the value is generally between −1 and 1). The closer the absolute value of Moran's I is to 1, the stronger the correlation is. When the index is 0, it indicates that the study objects are randomly distributed in space; when close to +1, it indicates the existence of spatial aggregation; and when close to −1, it indicates spatially discrete distribution. Thus, the spatial clustering model of facilities could be expressed [28]. In this study, the elderly population and the area of EDCCs in the residences were analyzed separately.

3.    Average nearest neighbor method (ANN)

An ANN can be used to describe the spatial distribution of a large number of points in a specific region, including three types of dispersed, random, and clustered distributions [29]. In this study, ANN analysis was conducted for the residences and EDCCs in Nankai District, respectively.

In ArcGIS, when the resulting R-value (ratio of ANN) is 1, the facilities are randomly distributed. If it is less than 1, the facilities are relatively close to each other in space and have a spatially clustered distribution. If it is greater than 1, the spatial distribution shows a dispersed or competing distribution pattern.

The returned Z-value and $p$-value were used to determine the statistical significance. Z-value represents the multiple of standard deviation (positive or negative). The higher the Z-value, the higher the degree of aggregation, and the closer to zero, the more random the distribution. $p$-value is the confidence interval, i.e., the probability of the observed spatial distribution pattern, which corresponds to the Z-value. When the absolute Z-value is larger, the corresponding $p$-value is smaller, and the facility is less likely to be randomly distributed in the space, and vice versa.

4.    Kernel density estimation method (KDE) and buffer zone method

A KDE is a spatial analysis in which the densities of elements in their domains are calculated and visualized. The method calculates the density of elements in the analysis of points around a single pixel, each of which has a smooth surface. The value is highest at the position of the point of interest (POI). As the distance from the POI location increases, the value gradually decreases and the connection with the facility decreases. The density distribution within the study area is then obtained by superimposing the densities at the

same locations and generating a gridded graph. Within the search radius $h$, the distance to other study objects is inversely proportional to the distance. KDE assumes that points in space are interconnected, and the strength of the connection is inversely proportional to the distance. This property is the same as the reality that the attractiveness of public service facilities decreases with distance [30].

KDE was performed for EDCCs at 300, 500, and 1000 m, respectively. Based on this, the proximity problem was solved by buffer zone analysis, which was used to explain the proximity or coverage of the facilities.

### 2.5. Accessibility Measurements

Most common measures of accessibility are evaluated from a regional or spatial perspective, with urban planning, transportation, and economic benefits as the objects of study. The different methods of measuring accessibility have their own applicable facilities and different usage situations.

1. Proportional method

The proportional method treats the study area as a "container," which can be a district, a sub-district, a community, a neighborhood, or a census unit, or can be subjectively defined according to the needs of the study. This method is often used to calculate the ratio of total service supply to total population in demand; higher ratios indicate better accessibility. The ratio method assumes that all facilities in the study area are only available to residents in the area. To make this assumption valid, the study area needs to be large enough to ensure that residents in the area would not choose facilities outside the area, and that residents outside the area would not choose facilities inside the area.

2. Nearest distance method

The nearest distance method, i.e., measuring the distance from the demand point to the nearest facility point, generally uses the Euclidean distance or road distance. This method assumes that residents will certainly choose the closest facility. It emphasizes the influence of distance on residents' activity decisions; the closer the distance, the better the accessibility. This method ignores the influence of other facilities with different levels, sizes, and types [31]. EDCCs serve elderly people, and the distance factor has a greater influence on their facility choice. Therefore, the nearest distance method has a certain reference value in the study. We obtained the accessibility values by applying the nearest facility point analysis of the Network Analyst module in ArcGIS. The accessibility expression was performed by the spatial interpolation method.

3. Two-step floating catchment area method (2SFCA)

The 2SFCA is essentially a combination of the floating catchment area method (FCA) and the gravity model, first proposed by Professors John Radke and Lan Mu at UCB and then improved by Luo and Wang [32,33]. The so-called "two-step floating" refers to two "movements" in the search area between the demand and supply points.

Step 1: Centering on supply point $j$, set the distance threshold $d_0$ to determine the supply point search area, search the set of demand points $k$ within the established search area, and calculate the supply-to-demand ratio $R_j$ for supply point $j$.

Step 2: With demand point $i$ as the center and $d_0$ as the radius, calculate the accessibility index $R_j$ of demand point $i$ in the search area and find the sum of supply and demand ratio $A_i$.

$$A_i = \sum_{j \in (d_{ij} \leq d_0)} R_j = \sum_{j \in (d_{ij} \leq d_0)} \frac{S_j}{\sum_{k \in (d_{kj} \leq d_0)} P_k} \tag{1}$$

where $A_i$ is the accessibility of demand point $i$. $R_j$ is the supply-to-demand ratio of supply point $j$ in the range of travel threshold $d_0$, i.e., the service capacity it provides. $S_j$ represents the supply scale of supply point $j$, which was quantified by floor area in this study. $d_{kj}$ is the distance between supply point $j$ and demand point $k$. $d_{ij}$ is the distance between point $i$

and point *j*. $P_k$ is the number of demand population in the search range. The accessibility of demand point *i* is positively correlated with the value of $A_i$.

Although the basic 2SFCA considers the supply–demand relationship between supply and demand points, it fails to address the effect of distance decay on accessibility. Therefore, to simulate a more realistic spatial relationship between supply and demand points, researchers have proposed to extend the 2SFCA with an additional distance decay function within the radius of the 2SFCA, whose general equation is:

$$A_i = \sum_{n=1}^{n} \frac{S_j f(d_{ij})}{\sum_{k=1}^{m} P_k f(d_{ij})} \text{ where } f(d_{ij}) = \begin{cases} f(d_{ij}), \ d_{ij} < d_0 \\ 0, \ d_{ij} > d_0 \end{cases} \tag{2}$$

After that, enhanced, gravity, kernel density, and Gaussian 2SFCA have been proposed and applied successively. The core operation is to use segmental assigned weights, distance decay function of gravity model, kernel density function, Gaussian function, etc., as distance decay function [34–37].

Referring to relevant Chinese studies on walking distance decay functions for the elderly in China, we chose the gravity 2SFCA (G2SFCA) as the measurement method:

$$f(d_{ij}) = d_{ij}^{-\beta}, \ d_{ij} \leq d_0 \tag{3}$$

where *β* is the decay parameter that takes a value generally between 1 and 2, which varies with the service type of the facility and the range of the study. EDCC is a non-emergency demand type facility, so it is generally taken as 1.

In this study, the "Origin–Destination (OD)" cost matrix was established in ArcGIS. The starting point is the residential site of the community (usually at the community gate or center of mass), the EDCC is the destination, the impedance is set as the length of the road, the limit travel distance is 300 m, the scale of the facility is the area of the EDCC, and the scale of the demand point is the number of households in the community.

When calculating the OD matrix, coordinate points overlap occurs because some EDCCs are built directly inside residential areas or transformed first floors of residential buildings. This overlap phenomenon is the ideal EDCC distribution pattern for a residential area. However, the OD distance score for such cases is 0, which cannot be calculated due to the issue of road network accuracy. To highlight the accessibility advantage of such situations, this study considered the field investigation and the accuracy of the road network, and set the OD cost distance of such situations as 10 m uniformly, which is the nearest walking distance from the residential unit to the facility.

4. Inverted 2SFCA (i2SFCA)

Wang designed a reverse 2SFCA method for evaluating facility congestion and supply coordination or saturation in 2017 [38]. By means of this method, this study tried to explain the saturation of EDCCs, i.e., the usage efficiency.

5. Potential model method

The potential model is an approach that introduces the law of gravity to urban economics and geography, revealing the relationship between economy and space in cities. The term "potential" refers to the potential energy of the relationship among objects in space. This energy is defined in the study of spatial accessibility as the potential for development, which decreases with distance. The potential model was first proposed and introduced for use by Hansen in 1959 with the following general expression for the model [4].

The original potential model only considers the potential effects of supply and demand points and ignores the countervailing effects of demand points. For example, population competition for the usage of facilities can cause the accessibility dilution. Alun E Joseph added the size factor $V_j$ to take into account the competing relationship between demand points [13].

$$A_i^p = \sum_{j=1}^{n} \frac{S_j}{d_{ij}^{\beta} V_j} \text{ where } V_j = \sum_{k=1}^{m} \frac{P_k}{d_{kj}^{\beta}} \tag{4}$$

where $A_i^p$ is the spatial accessibility from demand point $i$ to all facilities $j$. $S_j$ represents the scale of supply point $j$. $d_{ij}$ is the traffic impedance between node $i$ and node $j$, and $\beta$ is the friction coefficient, which takes the same value as above. $P_k$ is the size of the demand point, which was set as the population of the residence in this study. $n$ and $m$ denote the numbers of supply points $j$ and demand points $k$, respectively. $d_{kj}^{\beta}$ is the traffic impedance between demand point $k$ and supply point $j$.

Based on the characteristics of EDCCs and the research of other Chinese scholars, we added the service distance of elderly facilities to the potential model method for calculating the accessibility of elderly facilities. The facility service threshold distance was set to 300 m. The facility area was used as the size factor $V_j$. The transportation cost was the same as in 2SFCA, which was obtained through the OD cost matrix, and the value of $\beta$ was also taken as 1.

## 3. Results

### 3.1. Basic Statistics

#### 3.1.1. EDCC Statistics

The results of the EDCC data in Nankai District are shown in Table 5. It was found that the number of EDCCs in Nankai District is seriously insufficient, and there is no EDCC in some sub-districts. In addition, there are some discrepancies between the number of EDCCs in each sub-district and the distribution of the elderly population. For instance, the number of EDCCs in Shuishanggongyuan sub-district accounts for 21.74% of the total, but only 5.28% of the district's elderly requires an EDCC. Gulou sub-district, which has 5.42% of the district's residents and 1.27% of the elderly population, does not yet have an EDCC.

**Table 5.** Population ratio and EDCC data of Nankai sub-districts in 2018.

| Name | Number of EDCC | Ratio of EDCC (%) | Ratio of Elderly (%) | Ratio of Residence (%) |
|---|---|---|---|---|
| Gulou | 0 | 0.00% | 1.27% | 5.42% |
| Xingnan | 1 | 4.35% | 4.33% | 4.80% |
| Guangkai | 1 | 4.35% | 6.48% | 8.13% |
| Changhong | 1 | 4.35% | 6.05% | 5.58% |
| Xiangyanglu | 3 | 13.04% | 14.22% | 11.11% |
| Jialingdao | 2 | 8.70% | 11.56% | 8.57% |
| Wanxing | 3 | 13.04% | 15.36% | 13.55% |
| Xuefu | 2 | 8.70% | 8.60% | 7.58% |
| Shuishanggongyuan | 5 | 21.74% | 5.28% | 5.04% |
| Wangdingdi | 2 | 8.70% | 15.20% | 14.65% |
| Tiyuzhongxin | 1 | 4.35% | 5.52% | 11.73% |
| Huayuan | 2 | 8.70% | 6.13% | 3.85% |
| Total | 23 | 100.00% | 100.00% | 100.00% |

The "ratio" in each column refers to the ratio between their respective sub-district's value and the value in Nankai District.

#### 3.1.2. EDCC Density Statistics

According to the Nankai District Statistical Yearbook and the number of EDCCs in each sub-district, the density and proportion of EDCCs in each sub-district were calculated as shown in Table 6. The analysis results show that the number of EDCCs in each sub-district is seriously insufficient in terms of the capacity of households in 5 min pedestrian scale neighborhoods. The best situation is only 0.86 units per 4000 households in the Shuishanggongyuan sub-district. At the same time, there is a significant problem with the density and the number of facilities per 1000 elderly in each sub-district. Except for the Gulou sub-district, the sub-districts with relatively decent facility density are the Huayuan sub-district and the Shuishanggongyuan sub-district. The results were only 0.19 and 0.55 per 1000 elderly people, respectively. The highest average number of facilities in the

community was only 0.5 per community in the Shuishanggongyuan sub-district. The rest of the sub-districts had values of around 0.1.

**Table 6.** EDCC density statistics of Nankai sub-districts.

| Name | Density of EDCC ($/km^2$) | Average Number of EDCC (/4000 Households) | Average Number of EDCC (/1000 Elderly Over 60) | Average Number of EDCC (/Community) |
|---|---|---|---|---|
| Gulou | 0.00 | 0.00 | 0.00 | 0.00 |
| Xingnan | 0.59 | 0.18 | 0.14 | 0.11 |
| Guangkai | 0.59 | 0.11 | 0.09 | 0.08 |
| Changhong | 0.38 | 0.16 | 0.10 | 0.08 |
| Xiangyanglu | 0.30 | 0.23 | 0.12 | 0.16 |
| Jialingdao | 0.54 | 0.20 | 0.10 | 0.13 |
| Wanxing | 0.85 | 0.19 | 0.11 | 0.14 |
| Xuefu | 0.43 | 0.23 | 0.14 | 0.14 |
| Shuishanggongyuan | 0.89 | 0.86 | 0.55 | 0.50 |
| Wangdingdi | 0.40 | 0.12 | 0.08 | 0.09 |
| Tiyuzhongxin | 0.15 | 0.07 | 0.11 | 0.08 |
| Huayuan | 1.04 | 0.45 | 0.19 | 0.18 |

*3.2. Results of Spatial Distribution Analysis*

1. SDE

The corresponding SDE parameters (Table 7) obtained based on Section 2.4 are plotted and discussed in Section 4.

**Table 7.** SDE parameters of EDCCs and residences.

| Item | Center X | Center Y | X StdDist | Y StdDist | Rotation | Area ($km^2$) | Perimeter (km) |
|---|---|---|---|---|---|---|---|
| SDE of Residences | 117.144213 | 39.114371 | 0.031713 | 0.022796 | 162.990528 | 36.053588 | 22.544944 |
| SDE of EDCCs | 117.143797 | 39.107079 | 0.02796 | 0.020537 | 173.313796 | 28.634609 | 20.086912 |

2. SA

The results showed that the Z-value of residents and elderly population in Nankai District exceeded the critical value of 1.65 (rejecting the zero-dummy threshold). The *p*-value was less than 0.05, showing a significant positive autocorrelation, and a clustered distribution was identified. The Z-value of EDCC showed a negative autocorrelation. The *p*-value is greater than 0.1, indicating a high probability of a random distribution. This shows that the population spatial distribution effect was not considered in the preliminary planning, and there is a misplaced supply phenomenon (Figure 3).

3. ANN

The R-value of the community residences is less than 1, the absolute value of Z-value is greater than 1.65, and the *p*-value is less than 0.01. This indicates that their spatial distribution is relatively highly clustered, statistically significantly so (Figure 4a).

In contrast, the R-value of EDCC is greater than 1, the absolute value of Z-value is less than 1.65, and the *p*-value is greater than 0.01. It indicates that its spatial distribution presents a random distribution (Figure 4b).

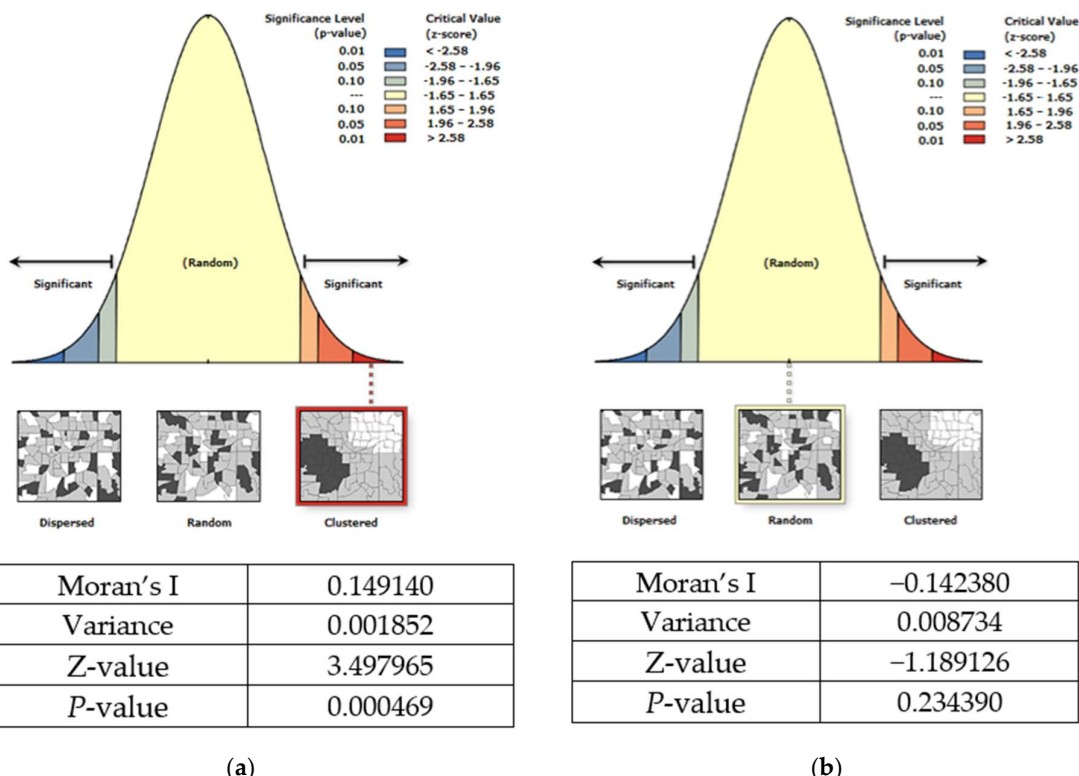

**Figure 3.** SA result: (**a**) elderly distribution; (**b**) EDCC distribution.

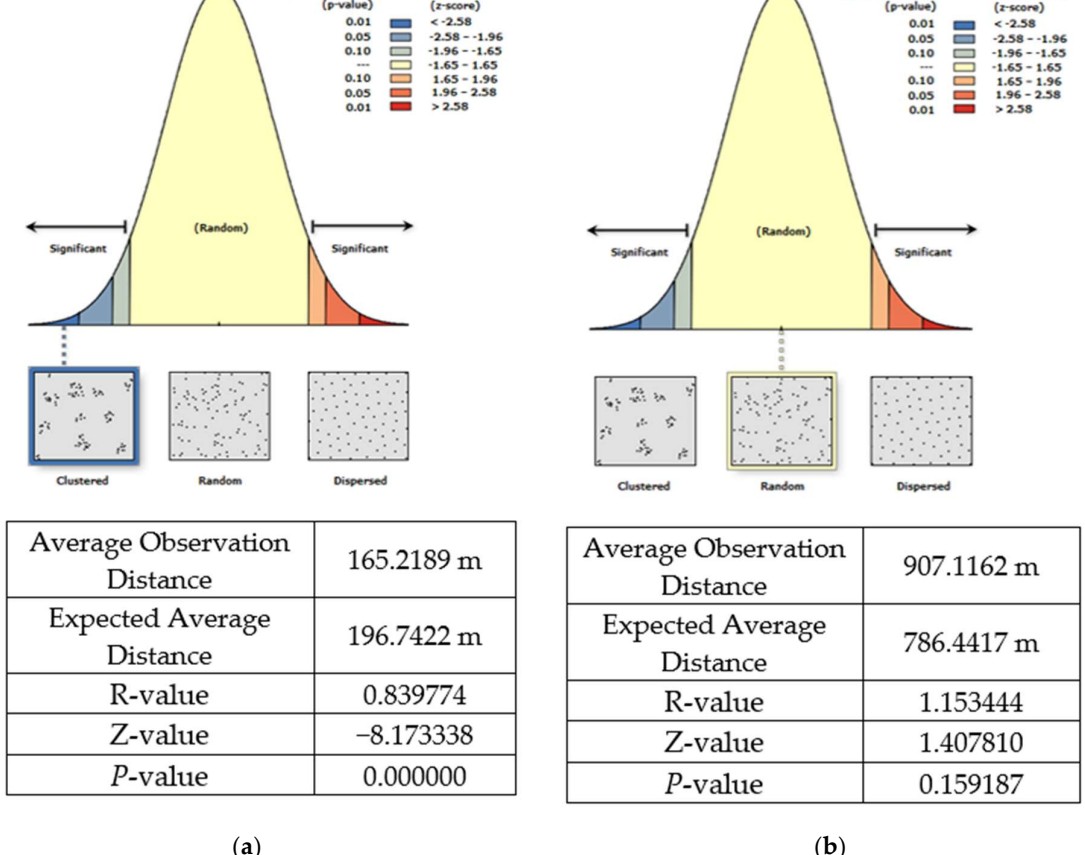

**Figure 4.** ANN result: (**a**) distribution of residences; (**b**) EDCC distribution.

4. KDE and buffer zone method

Comparing the kernel density heat maps of the elderly population obtained at different search distances (Figure 5), we can see that the distribution of EDCCs in Nankai District presents an irregular point distribution. There is no significant relationship between planning and the distribution of elderly population.

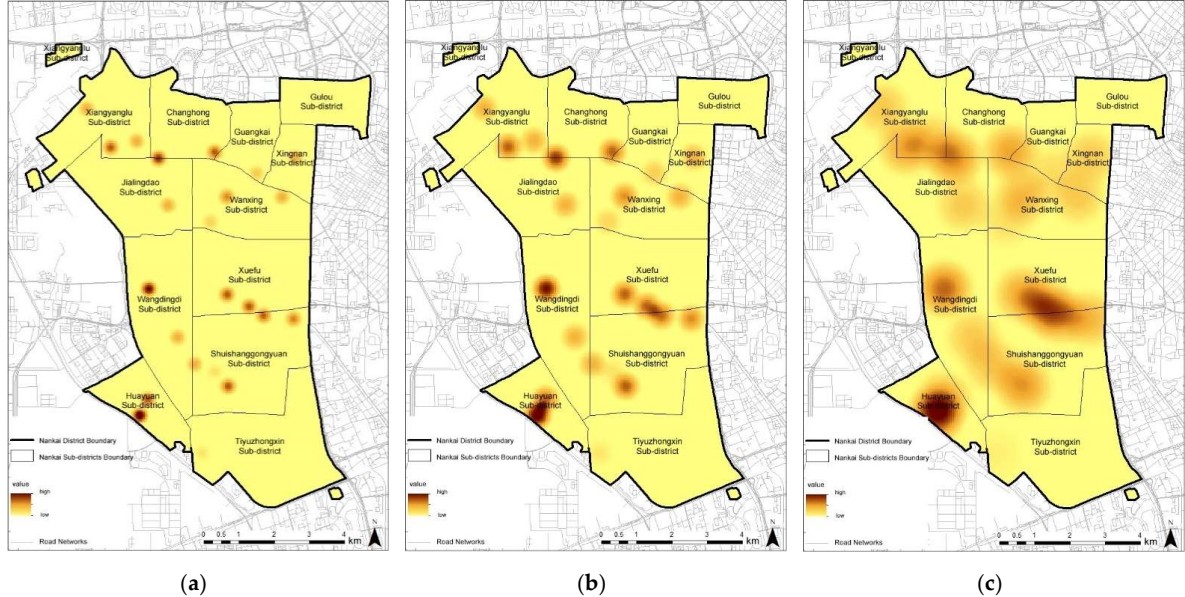

|   (**a**)   |   (**b**)   |   (**c**)   |

**Figure 5.** KDE of EDCCs: (**a**) 300 m radius; (**b**) 500 m radius; (**c**) 1000 m radius.

The areas with high densities of EDCCs are the Huayuan sub-district, the Wangdingdi sub-districts, and the junction of Shuishanggongyuan and Xuefu sub-districts. Most of the remaining sub-districts have missing facilities or very low densities. Regarding the existing results, the overall lack and unreasonable spatial distribution of EDCCs in Nankai District are the main problems.

The buffer zone analysis of the 23 existing EDCCs in Nankai District showed that none of community coverage, household coverage, and elderly population coverage exceeded 10%, and the facility coverage was extremely low (Figure 6).

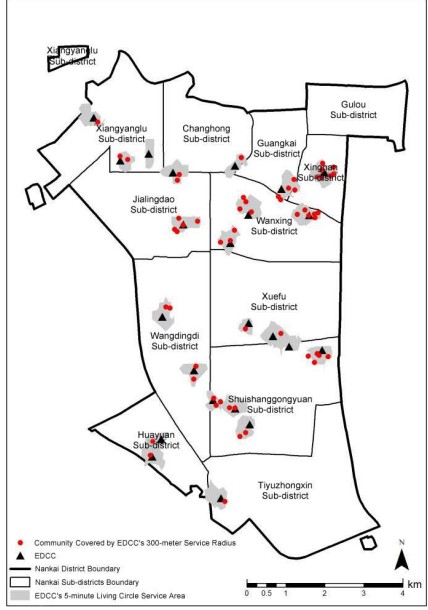

**Figure 6.** EDCCs in 300 m coverage area.

### 3.3. Results of Accessibility Measurements

1.  Proportional Method

The proportional method's accessibility values were obtained from the total service supply of the EDCCs in each sub-district divided by the number of elderly individuals in the sub-district. Subsequently, the data were prepared for visual display using the ArcGIS tool (Figure 7).

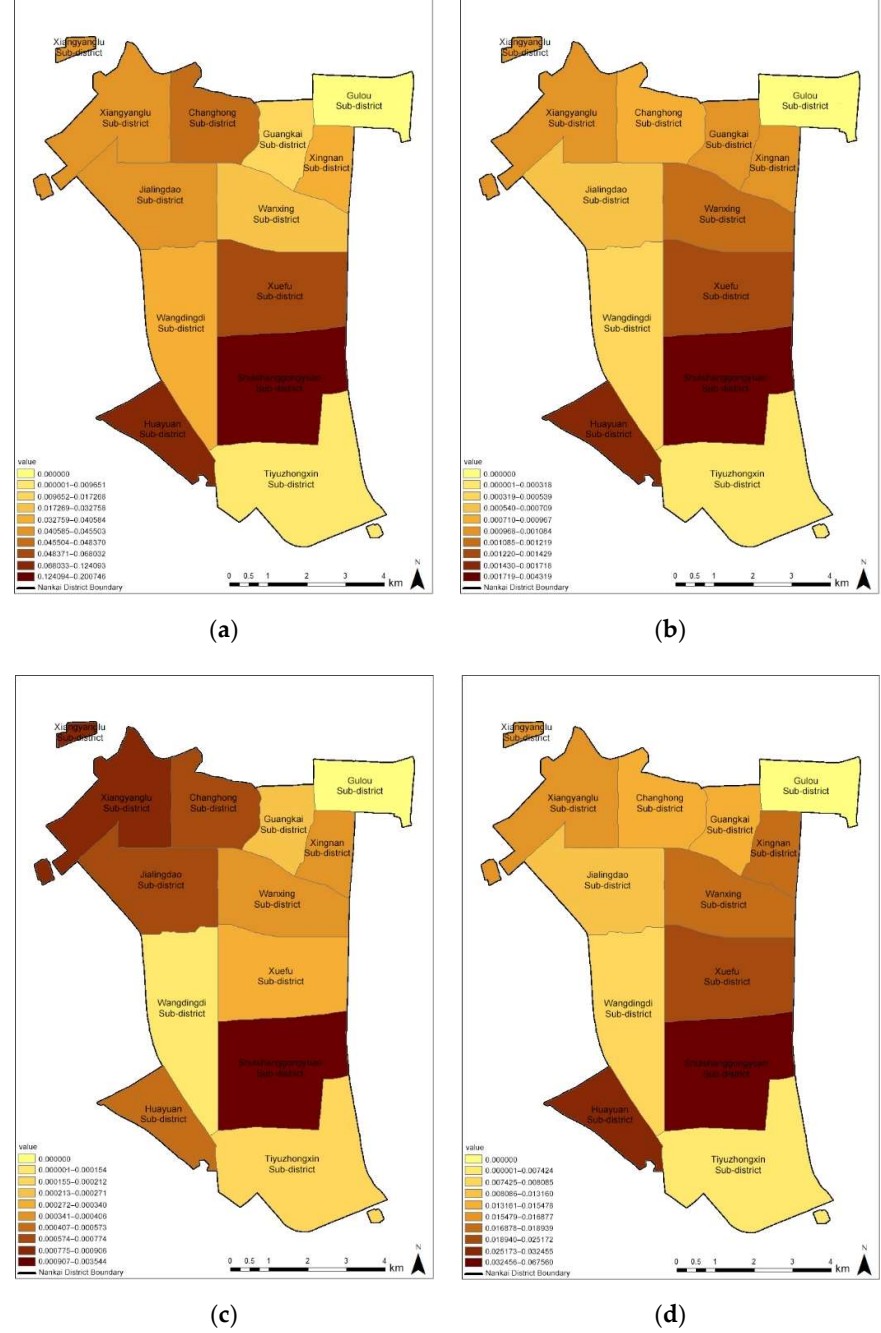

**Figure 7.** Proportional method results. (**a**) Facility area. (**b**) Number of beds. (**c**) Number of service staff. (**d**) Score of service.

2.  Nearest Distance Method

The results showed that 717 residences were generally far from the nearest EDCC (Figure 8). Only 49 residences were within 300 m (walking distance) of an EDCC. The

farthest distance among the 717 residences was 3493.17 m (Figure 9a). The shortest was the EDCC built directly within the block (Figure 9b).

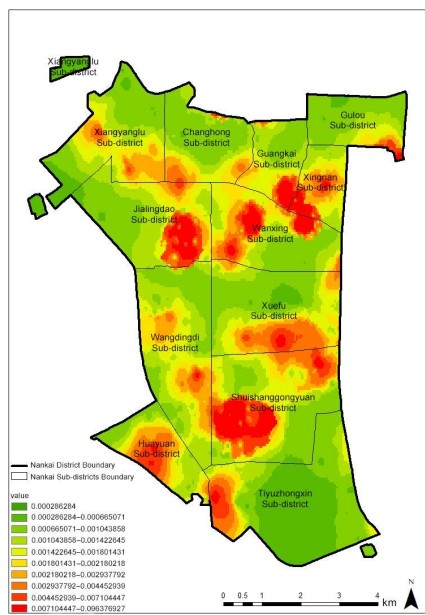

**Figure 8.** Nearest distance result.

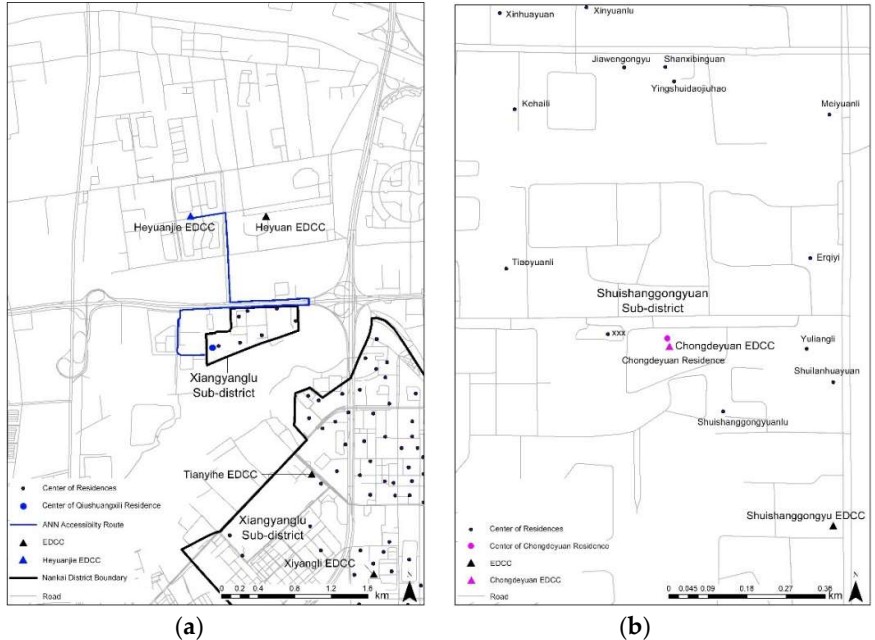

**Figure 9.** Two extreme cases of the nearest distance method. (**a**) Longest distance (3493.17 m from Qiushuangxili residence to Heyuanjie EDCC). (**b**) Shortest distance (Chongdeyuan EDCC built in Chongdeyuan residence).

3. i2SFCA

According to the residential area classification standards from the "Standard", the 5 min pedestrian scale neighborhood residential area is limited to a walking distance of 300 m. The number of dwellings is 1500–4000 units. The EDCC area was taken as the lower limit of 350 m². The number of dwellings was used as the upper and lower limits of the demand scale to calculate the EDCC saturation scores. The results showed the following EDCC saturation criteria: low saturation value of 4.28 and high value of 11.43.

Except for Santandongli community and Yangguangyibai community, the saturation was at the lowest value, i.e., in an inefficient supply state (Figure 10).

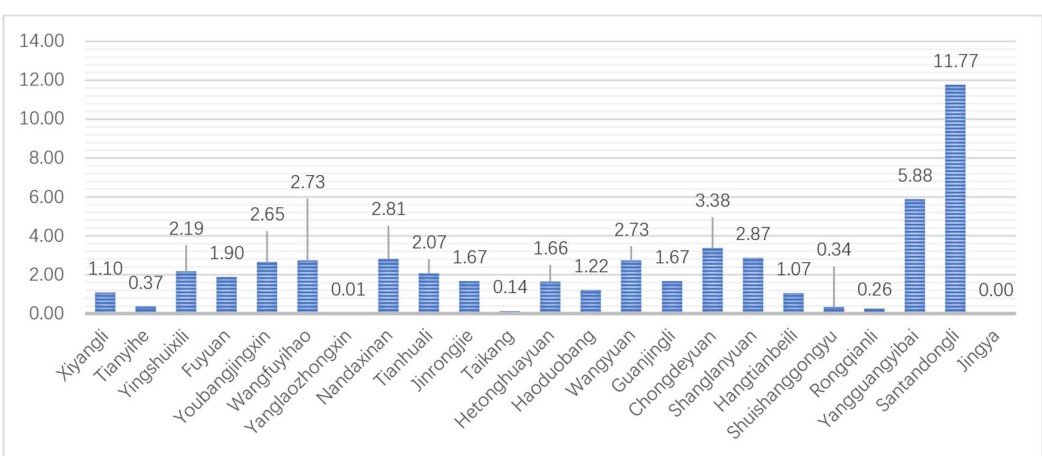

**Figure 10.** Saturation statistics of existing EDCC facilities in Nankai District.

## 4. G2SFCA and Potential Model Method

The results obtained by the G2SFCA and potential model methods are visually represented (Figure 11). The results show that the accessibility distributions presented by the two methods are very different.

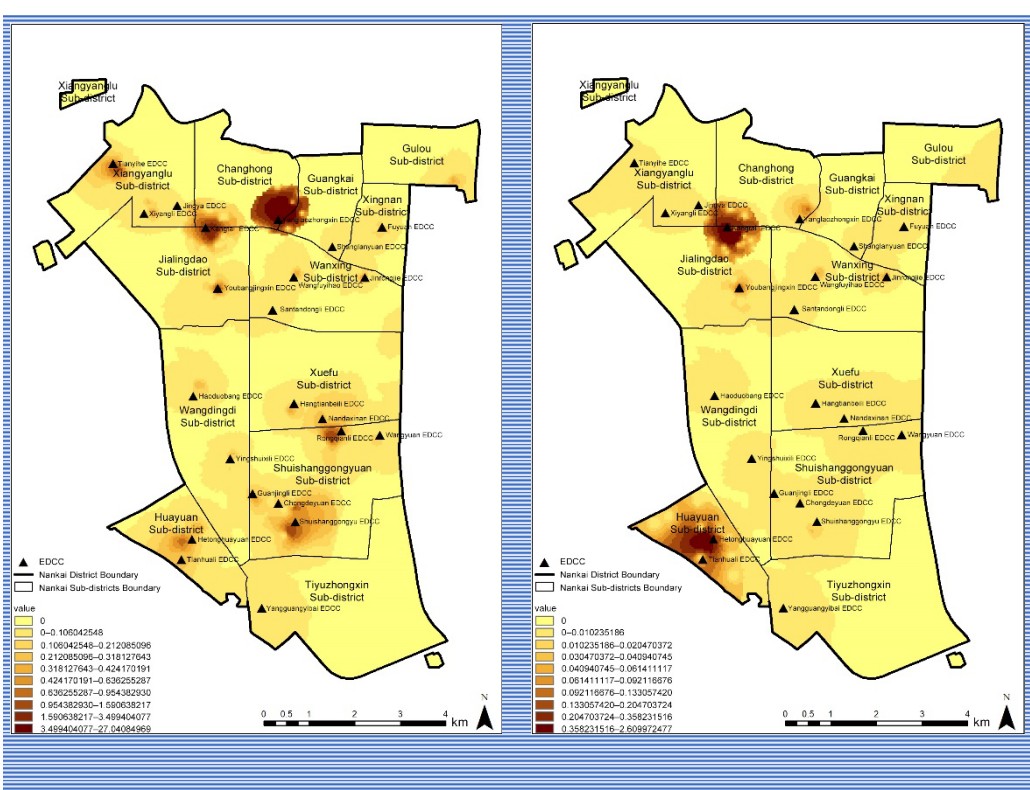

**Figure 11.** Comparison of differences in results. (**a**) Based on G2SFCA. (**b**) Based on potential model.

In the accessibility distribution map obtained by G2SFCA, the facilities with high accessibility are mainly concentrated in the Yanglaozhongxin, Kangtai, Shuishanggongyu, and Tianyihe sub-districts. Among them, the Yanglaozhongxin EDCC has the highest facility accessibility.

In the accessibility distribution map obtained by the potential model method with added service thresholds, the communities with higher accessibility are mainly concentrated in the Huayuan sub-district and the vicinity of the Kangtai EDCC. Among them, the accessibility of the neighborhoods in the Huayuan sub-district is the highest.

## 4. Discussion

Based on the results above, the following points should be considered in terms of the spatial distribution and accessibility of EDCCs in Nankai District. This could serve as advice for other Chinese urban built-up areas.

1.  Significant quantity shortage

The number of existing EDCCs in Nankai District is seriously low. The existing 23 EDCCs are far from the minimum number of allocations required by the Standard. Most of the communities are not served by an EDCC. The accessibility measurement shows that only the residences around the EDCCs have accessibility values. Less than 10% of the community has EDCC coverage.

The analysis results show that the accessibility of Shuishanggongyuan and Huayuan sub-districts is higher than that of other sub-districts. Gulou sub-district has the lowest accessibility in the entire district. Other sub-districts showed no significant changes in the accessibility results for the four indicators during the analysis. Xiangyanglu and Jialingdao sub-districts showed increased accessibility in the measurement of the population served. The reason for this is that the EDCCs in these two sub-districts are mostly installed in nursing homes, and the number of served is calculated as in institutional nursing facilities. This is one of the advantages of the joint construction of EDCC and institutional elderly care centers.

According to the field investigation, two other phenomena were found to exist. On the one hand, there are several EDCCs under construction or undergoing validation. To ensure the quality of facility services, some private facilities with EDCC functions have not yet been accepted for entry into the relevant subsidy and management lists. The improvement of these facilities may meet a part of the demand in the future. On the other hand, the demand for "partial functions" of the existing EDCCs has already been satisfied. For example, the number of elderly canteens in Nankai District has reached 185, with an official claim of 100% community coverage. However, according to the data of Lianjia, there are about 460,000 households in Nankai District. At least 112 EDCCs are needed for every 4000 households. Therefore, the existing facilities can rarely meet the demand both in coverage and quantity.

2.  Spatial misplacement of supply

The SDE results showed that the spatial distribution of EDCC in Nankai District presented spatial misplacement with the distribution of elderly population. By comparing the SDE parameter tables generated by the calculation (Table 7), the SDE flat rate of EDCCs is larger than that of residences. The direction of EDCCs distribution is obvious, and the degree of dispersion is larger. By comparing the SDE areas of them, the area of residences is larger than that of EDCCs, indicating that the distribution of facilities does not cover all residences (Figure 12).

Therefore, it could be presumed that Nankai District missed the preliminary spatial distribution planning for EDCCs. Through interviews with the person in charge of the Elderly Services Division of the Tianjin Civil Affairs Bureau, we learned that the existing EDCC construction model in Tianjin is that the sub-district proposes construction requirements, the District Civil Affairs Bureau approves and submits them, and finally, the Municipal Civil Affairs Bureau allocates funds or subsidizes the construction. In terms of the construction of facilities, the idea of construction is mostly a matter of: "Where could it be available? Where can it be built? Where must it be built?" Due to the historical reasons for urban built-up area, the construction of facilities was preferred to the financial power,

land property, and the nature of the residence (whether it is a high-class neighborhood or a retirement community), with less consideration for the spatial coupling with the elderly.

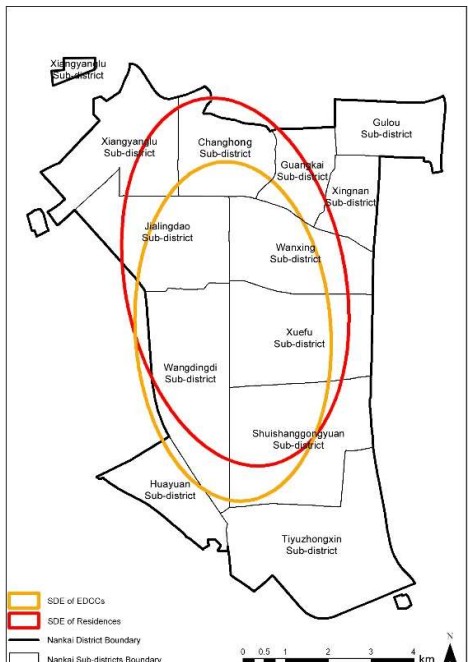

**Figure 12.** SDE comparison of EDCC and elderly population spatial distribution.

This will inevitably pose both equity and efficiency deficits to the operation of ED-CCs. In terms of equity, the accessibility analysis shows that most communities are in an inaccessible state, and some communities are over-accessible (Figure 11). In terms of efficiency, the saturation analysis shows that the saturation of EDCC usage appears to be over-saturated in some cases at 11.43, while in others it appears to be unavailable at 0, as the distance threshold is exceeded (Figure 10). The above indicate that: the accessibility of facility distribution is uneven, i.e., inequitable; the saturation of facility usage varies greatly, resulting in inefficiency of facility usage.

3. Service distance requires further verification

According to the Standard, the saturation of existing EDCC usage analyzed by i2SFCA is significantly lower than the target under the 300 m service distance threshold. The attainment rate is less than 10%, and most of the EDCC is "unreachable". This result is significantly different from the actual investigation. Therefore, it is necessary to further analyze the reasons behind this, and re-calculate the appropriate service distance thresholds considering the maximum efficiency of facility use and the walkability of the elderly.

4. The potential model method is more suitable for EDCC accessibility analysis

When comparing the equations of G2SFCA and the potential model, it can be found that the distance decay function $f(d_{ij})$ is included in both the numerator and denominator of the G2SFCA equation, whereas the potential model equation only includes the transportation cost factor $d_{ij}{}^{\beta}$ in the denominator. Due to the small distance threshold and small number of existing EDCC services in the urban built-up areas of China, this result is a one-to-one correspondence between the supply and demand points in most cases (Figure 13). This then leads to the G2SFCA equation being approximately reduced into:

$$A_i = \frac{S_j}{P_k} \tag{5}$$

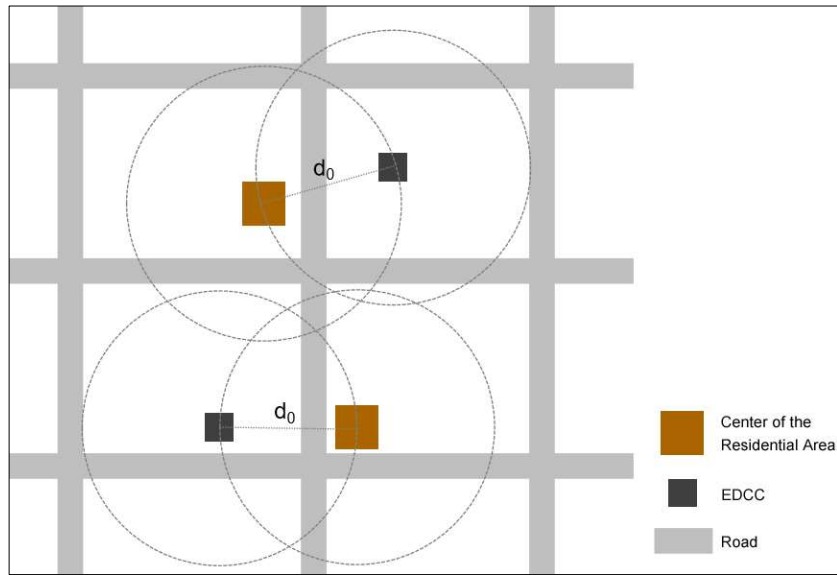

**Figure 13.** The one-to-one correspondence between supply and demand points within 2SFCA.

In this case, the equation no longer considers the facility traffic cost, but only the supply and demand. The potential model method, on the other hand, still considers the distance decay of transportation costs in such cases. This leads to differences in the calculation results of the two methods for the accessibility measure. According to the results of the field study, the potential model method with distance threshold can more accurately simulate the real travel and the usage of the elderly in the calculation of the accessibility of EDCCs in China's urban built-up areas.

5. The principles of planning and optimizing for EDCCs

Through the previous analysis of the construction process and distribution characteristics of EDCCs in urban built-up areas, the following strategies and objectives should be considered in further planning and distribution optimization.

(1) Aiming for community-wide accessible.

The deteriorating mobility habits and abilities of the elderly dictate that their activities are concentrated in community areas. This is particularly evident in built-up urban areas with high building density. Therefore, EDCCs should be within pedestrian accessible range of the elderly in the corresponding communities to ensure their equity.

(2) Indicating with the demand of residents.

The scale of EDCC should be coupling verified according to the elderly population within its service distance. This will improve the balance and efficiency of its spatial distribution and enable the integrated and coordinated development of the regional community elderly service system.

(3) Optimizing by progressive phased construction.

Firstly, the reasonable service distance threshold of EDCC in the region should be determined based on data analysis and simulation to maximize its utility. Secondly, the service distance can be gradually reduced, such as from 500 to 300 m, to guide the progressive phased construction with comprehensive consideration of cost and requirements.

(4) Composing based on the existing buildings.

Insufficient dedicated facilities and land constraints are commonly found in built-up areas of Chinese cities. When carrying out EDCC allocation in such areas, existing buildings, such as elderly institutions, community councils, community health service stations, or residences, can be considered for composite construction methods. These can be used as the candidate points for phased optimization and construction.

6.    Limitations

This study attempted to interpret the arrangement of EDCCs in the urban built-up area through spatial distribution and accessibility analysis, relying only on the corresponding standards of community living circle. With this aim, we tried to propose a planning philosophy for the allocation planning and construction of EDCC in the area. However, the spatial arrangement is only one of the important factors affecting the construction of community elderly service system and the actual utilization of the elderly. In practice, constrained by social, economic, cultural, and medical factors in the built-up area, further evidence from a wider scope is still needed, while also relying on constant improvement and optimization in operation.

In addition, the development of advanced technologies such as smart wear and service-oriented robots has also led to more possibilities and alterations in architecture design and planning arrangement. Through shared services, machine supervised services, and others, the physical space requirements and accessibility of EDCCs may produce a new definition, which also bring more opportunities and challenges for future research.

## 5. Conclusions

This study took 12 sub-districts, 171 communities, 717 residences, and 23 EDCCs in Nankai District of Tianjin as the research objects and evaluated the spatial distribution and accessibility status of EDCCs in this region through calculations by relevant technologies and algorithms of the ArcGIS platform. The main conclusions of the study are as follows.

1.    Different spatial distribution analysis methods have their own emphases and limitations. In practice, multiple methods need to be applied to obtain the spatial distribution characteristics of facilities in a comprehensive manner. The Moran's I can be used to analyze the dispersion, randomness, and clustering of the spatial distribution of facilities, but the degree of clustering or dispersion of facilities needs to be analyzed by the ANN. The ANN can only analyze the degree of clustering of facilities in space, but cannot accurately analyze the distribution area and form of facilities. The KDE can form a smooth space of discrete degree facility distribution and visualize the density analysis map, thereby making up for the limitation of the ANN.

2.    The goal of conventional accessibility models is to illustrate the ratio between facility supply capacity and demand, and the spatial relationship between them. This can be used to evaluate the accessibility of public services to residents. The i2SFCA can be used to capture the "crowding" of facilities, i.e., the scarcity of facilities or the intensity of competition. However, the potential modeling approach is better suited to simulating the actual accessibility of such facilities that are limited in number, insufficient in supply, and have a small service distance, and where distance has a greater impact on residents' choices.

3.    The Standard implemented in 2018 proposes that the service distance of an EDCC should not exceed 300 m. Through actual investigation, simulation, and calculation, it was found that there is a low utilization rate of facilities, wastage of resources, substandard saturation, and even no residences in part of the service distance, according to the limitation of a 300 m service distance threshold. It is necessary to further study whether the service distance can be set to 500 m or others, as in a transitional construction scheme.

4.    Through spatial visualization and statistics of the 6th Census data and residence data, it was found that: the regional distribution of the elderly population in Nankai District varies significantly in each sub-district and has the characteristics of clustering distribution. The distribution of the elderly population shows a trend of high in the west and north, and low in the east and south. The EDCCs in the region have the problems of random spatial distributed, mismatched supply, and insufficient quantity. The problem of low saturation of facility services within the service distance exists also.

The problem of community aging in place has plagued China and countries around the world for a long time. The construction of a sophisticated elderly service system involves social, economic, cultural, medical, urban planning, and architectural factors. This study only analyzed the corresponding spatial distribution and accessibility from the perspectives of architecture and urban planning. However, architecture and spatial arrangement are only one of the factors that influence the construction of community elderly service systems and the actual decisions of elderly people to use them. In the practical planning, multi-disciplinary, multi-contextual, and multi-angle argumentation should be performed. At the same time, it needs to be constantly improved and optimized in use. Only then can the EDCC be truly built to the satisfaction of the elderly and the acceptance of the residents.

Based on this study, we will further explore the EDCC service distance thresholds that are suitable for urban built-up areas by combining relevant standards, population data, and elderly travel modes. This will be used to identify construction strategies and candidate sites. Secondly, a stepwise optimization of site selection for new EDCCs in Nankai District will be carried out by using the "regional allocation model." After that, the new sites will be evaluated and optimized mainly by the potential model method. Finally, the feasibility of the proposal will be evaluated with the practical investigation.

**Author Contributions:** Conceptualization, D.W. and H.F.; methodology, S.Z. and H.L.; software, J.G.; validation, D.W., L.G. and P.L.; formal analysis, D.W.; investigation, J.G. and D.Q.; resources, S.Z. and H.L.; data curation, D.W. and J.G.; writing—original draft preparation, D.W. and J.G; writing—review and editing, D.W., D.Q. and H.F.; visualization, J.G. and L.G.; supervision, S.Z.; project administration, D.W. and H.F.; funding acquisition, H.L. and P.L. All authors have read and agreed to the published version of the manuscript.

**Funding:** This work was supported by the National Key Research and Development Program of China (grant number 2019YFD1100402) and the Key Technology Research and Development Program of Shandong (grant number 2019GSF110004).

**Institutional Review Board Statement:** Not applicable.

**Informed Consent Statement:** Not applicable.

**Data Availability Statement:** Data available on request due to restrictions, e.g., privacy or ethics.

**Acknowledgments:** We are very grateful to Yu Miao (Department of Civil and Environmental Engineering, University of Los Angeles, CA, USA) for his generous advice and help in the writing process, and the DAMlab (Digital Architecture and Manufacture Laboratory) Tianjin Sector at Tianjin Chengjian University for providing excellent equipment support during this study.

**Conflicts of Interest:** The authors declare no conflict of interest.

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
