# Peer review of "Spatial Distribution and Accessibility Measurements for Elderly Day Care Centers in China’s Urban Built-up Area: The Case of Tianjin Nankai District"

_buildings, doi:10.3390/buildings12091413_

Round 1

Reviewer 1 Report

 The article is quite interesting for researchers who are interested in methods as it  try “ to provide a theoretical basis and methodological support for the  planning and renovation of EDCCs in China’s old urban areas by examining the spatial distribution and accessibility measurement methods (l.77-79). On the basis of  Tianjin Nankai District the Authors asses spatial accessibility for older people. Some result of the investigation are not surprising as in many old urban areas “there is no significant relationship between planning and the distribution of elderly population”. It would be worth to notice that old urban areas have their limitations.

 Shortages:

·    *It was noted that the idea of Chinese elderly community day care model is “an emergent solution to the aging problem influenced by the Eastern perspective of family in China” (l.14-15; 38-39). For readers outside of China it is insufficient information what is the specificity of this idea. The only one sentence (lines 42-43) doesn’t  make change. It would be nice if the Authors try to compare with the possible solutions existing in other countries.

·      * L.45 “community planning system of "per 1000 population 45 model" in China focused on supply-side planning”. More clarifications are needed.

·        * It was mentioned that “the built-up China’s old urban areas have different traffic organization, building status, and population composition”(l.69-70). It should be explained what does it mean, we have a lot of old urban areas in Europe, Asia. What is – if it does – the specificity of China’s old urban areas.

·         *The Authors underline the “uniqueness of China's community aging mode” – more information is needed.

·        * The Authors concentrate on Tianjin Nankai District, they describe the demographic situation of that area giving us some data on ageing and the older people living in that district. It would be great if they give us possibility to have a glance on the ageing situation of other districts. Why? Because we don’t know if Tianjin Nankai is typical one if we talk about the ageing; China is generally ageing very fast – give us some data.

·       *  L.113-114: “2017, the registered population of the district is 859.298 million, of which 243.43 million are aged 60 or above, accounting for 28.33% of the  total registered population of the district. In 2017, there were 859,298 people registered in (…)” What number is correct? Are we talking about millions or thousands.

·         *It would be better if the Authors state that “aging rate” is the percentage of people 60 and over (table1).

·        * Title tab. 3 - add please that “Elderly population of Nankai Sub-districts” in 2010.; Table 5. “Population and EDCC data of Nankai Sub-districts” …in which year?

·         *Table 5. How do you calculate the ratios? Ratio of EDCC (%) Ratio of Elderly (%) Ratio of Residence (%); give explanations.

·     *Table 6. It would be more clear if you decide to write Average Number of EDCC/4000 Households; Average Number of EDCC/No of people 60? Or 65?) etc.

Author Response

Dear reviewer,
Thank you for your constructive comments on our manuscript entitled "Spatial Distribution and Accessibility Measurement of Elderly Day Care Centers in China’s Old Urban Areas: The Case of Tianjin Nankai District" (ID:buildings-1868953). Those comments are very helpful for revising and improving our paper, as well as the important guiding significance to further research. We have studied the comments carefully and made corrections which we hope meet with approval. The main corrections are in the manuscript and the responses to your comments are as follows.

Revise "old urban areas" in the title to "urban built-up areas" to make it more rigorous.
Attempt to clearly address the practical significance of EDCC as a model of elderly care in the context of traditional Chinese family concepts.
Explain the typicality of Nankai in terms of elderly care in China.
Reviewed thoroughly, especially in terms of data and tables, for rigor and accuracy.

A detailed description of the revisions to your review comments can be found in the attached file.
We appreciate your valuable comments and hope that the correction will meet with your approval.
Looking forward to hearing from you.

Yours sincerely,
Da Wan

Reviewer 2 Report

Thank you for giving me this opportunity to read the manuscript entitled "Spatial Distribution and Accessibility Measurement of Elderly Day Care Centers in China's Old Urban Areas: The Case of Tianjin Nankai District". The topic of this manuscript is interesting and would be a good contribution to this field. I think it could be considered for publication in Buildings once the following issues are addressed.

1.     Please replace the keywords that already appear in the manuscript's title with close synonyms or other keywords, which will also facilitate your paper to be searched by potential readers.

2.     Line 48: "…, the latest "Urban Residential Area Planning and Design Standard (GB 50180-2018)": reference should be added here.

3.     Line 135: Baidu Map API is just an API instead of a dataset. The authors should clarify the description here.

4.     It is good to see that the authors used Baidu Map API to conduct the analysis; however, detailed information is never provided in the manuscript about what and how the spatial distribution and accessibility analysis is performing based on Baidu Map API.

5.     Line 135: OpenStreetMap is known to have detailed and complex road information. In addition to the different levels of roads, single and dual-lane problems in the original dataset also need to be identified and addressed prior to the accessibility analysis. The authors should give detailed information regarding how the OSM road network was used in the analysis in this manuscript.

6.     In the urban areas, especially high-density urban areas like Nankai District of Tianjin, the distribution of residents or elderly population has significant spatial heterogeneity. I think the authors would also agree that the population distribution characteristics presented in Figure 3 are unconvincing to the actual situation. I would therefore like the authors to justify the usage of the kernel density approach to generate spatially continuous population distribution maps.

7.     Line 57-59 "Regarding the spatial distribution of public service facilities, the main research objects involve high demand for physical proximity of urban public service facilities such as schools, hospitals, and parks [7–10].": Some newly published papers could be cited here to support the statements here, for example, the papers titled "Dynamic assessments of population exposure to urban greenspace using multi-source big data" and "How does urban expansion impact people's exposure to green environments? A comparative study of 290 Chinese cities".

8.     Section 2.4 and 2.5: Excessively redundant information about methods needs to be removed. Most of the methods just need a brief description of functions and the corresponding references, as they are not developed or modified by the authors.

9.     The authors mentioned using a software named "ArchGIS", but I think it should be "ArcGIS".

10.  "Limitation" should be added as a sub-section of "Discussion".

11.  Some grammatical errors exist in the manuscript. Therefore, a critical review of the manuscript language will improve readability.

Author Response

Dear reviewer,
Thank you for your constructive comments on our manuscript entitled "Spatial Distribution and Accessibility Measurement of Elderly Day Care Centers in China’s Old Urban Areas: The Case of Tianjin Nankai District" (ID:buildings-1868953). 
We are honored to receive your professional and insightful review! Those comments are very helpful for revising and improving our paper, as well as the important guiding significance to further research. We have studied the comments carefully and made corrections which we hope to meet with approval. The main corrections are in the manuscript and the responses to your comments are as follows.

Revise "old urban areas" in the title to "urban built-up areas" to make it more rigorous.
Attempt to clarify the relevant data acquisition and conversion methods.
Simplified and removed some unhelpful or redundant parts to improve the efficiency of reading while increasing the rigor of the manuscript.
Reviewed thoroughly, especially in terms of data and tables, for rigor and accuracy.
Improve the English expressions through native editing.

A detailed description of the revisions to your review comments can be found in the attached file.
We appreciate such valuable comments and hope that the correction will meet with your approval.
Looking forward to hearing from you.

Yours sincerely,
Da Wan

Reviewer 3 Report

Thank you for the opportunity of reviewing the interesting paper. It covers the important issue of distribution and accessibility of elderly day care centers in China.  I have the following comments:

 ‐ In discussion, please provide analysis from the perspective of the equality and efficiency as abstract describe in line 27.

 - The approaches of planning for elderly day care centers should be proposed by using these analyzing methodologies.

 - Line 502 “more than 90% of the communities are in the accessible state.” Please provide evidence of supporting this figure.

Author Response

Dear reviewer,
Thank you for your constructive comments on our manuscript entitled "Spatial Distribution and Accessibility Measurement of Elderly Day Care Centers in China’s Old Urban Areas: The Case of Tianjin Nankai District" (ID:buildings-1868953). Those comments are very helpful for revising and improving our paper, as well as the important guiding significance to further research. We have studied the comments carefully and made corrections which we hope to meet with approval. The main corrections are in the manuscript and the responses to your comments are as follows.

Revise "old urban areas" in the title to "urban built-up areas" to make it more rigorous.
Detailed discussion of equity and efficiency is carried out.
Discussion of strategies and principles related to EDCC planning and optimization has been added.
Clarified a misunderstanding caused by inaccurate language.
Reviewed thoroughly, especially in terms of data and tables, for rigor and accuracy.

A detailed description of the revisions to your review comments can be found in the attached file.
We appreciate your valuable comments and hope that the correction will meet with your approval.
Looking forward to hearing from you.

Yours sincerely,
Da Wan

Round 2

Reviewer 2 Report

Thank you for giving me this opportunity to read the revised version of the manuscript titled "Spatial Distribution and Accessibility Measurement of Elderly Day Care Centers in China’s Urban Built-up Area: The Case of Tianjin Nankai District", and for the detailed responses to my earlier comments. I am satisfied with this revised version, and I think it is acceptable now.